

# β-Defensin *versus* conventional markers of inflammation in periprosthetic joint infection: a retrospective study

Javier Fernández-Torres[1,2,*], Yessica Zamudio-Cuevas[1,*],
Karina Martínez-Flores[1], Ambar López-Macay[1], Graciela Rosas-Alquicira[3],
María Guadalupe Martínez-Zavaleta[3], Luis Esaú López-Jácome[3],
Rafael Franco-Cendejas[4] and Ernesto Roldan-Valadez[5]

[1] Laboratorio de Líquido Sinovial, Instituto Nacional de Rehabilitación Luis Guillermo Ibarra Ibarra, Mexico City, Mexico
[2] Departamento de Biología, Facultad de Química, Universidad Nacional Autónoma de México, Mexico, Mexico
[3] Clinical Microbiology Laboratory, Instituto Nacional de Rehabilitación Luis Guillermo Ibarra Ibarra, Mexico City, Mexico
[4] Biomedical Research Subdirection, Instituto Nacional de Rehabilitación Luis Guillermo Ibarra Ibarra, Mexico City, Mexico
[5] División de Investigación, Instituto Nacional de Rehabilitación Luis Guillermo Ibarra Ibarra, Mexico City, Mexico
[*] These authors contributed equally to this work.

Corresponding author
Javier Fernández-Torres, javierastrofan1971@gmail.com

## ABSTRACT

**Background**. Diagnosing periprosthetic joint infection (PJI) remains a significant challenge for healthcare professionals. Commonly utilized inflammatory markers include erythrocyte sedimentation rate (ESR), C-reactive protein (CRP), and white blood cells (WBC). Human β-defensin 1 (β-defensin) is an antimicrobial peptide elevated in infection, yet its diagnostic value for PJI has not been explored. The purpose of this study was to evaluate the efficacy of synovial β-defensin as a diagnostic marker for PJI and to compare its performance with ESR, serum CRP, and WBC.

**Methods**. We conducted a single-center retrospective study from October 2022 to June 2023. A total of 105 joint fluid samples from revision patients at the Instituto Nacional de Rehabilitación Luis Guillermo Ibarra Ibarra were collected intraoperatively (71 hips, 34 knees) and frozen. According to MSIS criteria, 64 patients were defined as positive for PJI and the remaining 41 were negative. Synovial β-defensin levels were quantified using ELISA, serum CRP levels by immunoturbidimetry, and blood ESR and WBC were analyzed. Sensitivity and specificity were determined using ROC curves, and diagnostic performance was compared using the area under the curve (AUC). Cut-off values for diagnosing PJI were established.

**Results**. Levels of synovial β-defensin, ESR, serum CRP, and WBC were significantly higher in the PJI group compared to the non-PJI ($P < 0.0001$). The AUCs were 0.948 for β-defensin, 0.884 for ESR, 0.902 for CRP, and 0.767 for WBC, with a combined AUC of 0.994. Sensitivity/specificity for β-defensin, ESR, CRP, and WBC were 0.966/0.830, 0.887/0.791, 0.930/0.771, and 0.820/0.682, respectively. Optimal predictive cut-off values were 1105.8 pg/mL for β-defensin, 11.5 mm/h for ESR, 5.55 mg/L for CRP, and $7.3 \times 10^3/\text{mm}^3$ for WBC.

**Conclusion**. The synovial $\beta$-defensin assay demonstrated greater sensitivity and specificity for the diagnosis of PJI compared to ESR, serum CRP and WBC. Therefore, $\beta$-defensin shows promise as a diagnostic marker for PJI. Simultaneous determination of all markers may increase diagnostic confidence.

## INTRODUCTION

Periprosthetic joint infection (PJI) remains one of the most significant challenges in orthopedic surgery (*Nelson et al., 2023*). Infection (25.2%) and implant loosening (16.1%) are the most common causes of revision total knee arthroplasty (*Bozic et al., 2010*; *Bozic et al., 2009*). Timely and accurate diagnosis is crucial for the successful treatment of PJI with implant retention. Due to the nonspecific nature of pain, a common symptom of PJI, various tests have been employed to differentiate between septic and aseptic etiologies for revision surgery, yielding mixed results. Coagulase-negative staphylococci, primarily *Staphylococcus epidermidis* (*S. epidermidis*), often form tightly adherent biofilms on implants, exhibiting high antibiotic resistance and low microbiologic detection rates (*Benito et al., 2019*; *Banke et al., 2020*).

Diagnosing PJI involves distinguishing between septic and aseptic processes using various methods. Quantifying markers such as erythrocyte sedimentation rate (ESR), serum C-reactive protein (CRP), white blood cell (WBC) count, and microbiological culture is essential (*Deirmengian et al., 2014*). Microbiological culture remains a cornerstone due to its high sensitivity and specificity, crucial for guiding appropriate antimicrobial treatment (*Jordan et al., 2014*). Given the complexity of PJI detection, additional methods like implant sonication (*Trampuz et al., 2007*), molecular techniques (*Bergin et al., 2010*), and interleukin-6 (IL-6) quantification (*Di Cesare et al., 2005*) have been explored, though they significantly increase costs.

Among systemic markers, serum CRP is widely used to indicate PJI, as its levels rise in response to infection and help evaluate treatment efficacy and infection resolution. However, serum CRP is not specific to infection and may be elevated in other inflammatory conditions such as rheumatoid arthritis or autoimmune diseases (*Ghanem et al., 2009*; *Tetreault et al., 2014*). Therefore, PJI diagnosis typically involves a combination of laboratory markers. In a recent study by *Moldovan (2024)*, new biomarkers for the diagnosis of PJI, such as serum neutrophil lymphocyte ratio (NLR), serum platelet lymphocyte ratio (PLR), serum monocyte lymphocyte ratio (MLR), serum systemic inflammation index (SII), serum systemic inflammatory response index (SIRI), and serum monocyte lymphocyte ratio (MLR) were evaluated. Data from this study suggest that SII with cutoff value >605.31 and NLR with cutoff value >2.63 may increase the diagnostic accuracy of PJI when used in conjunction with other established parameters (*Moldovan, 2024*).

Antimicrobial peptides play vital roles in defending against microorganisms. They exert direct cytotoxic effects on bacteria, fungi, parasites, and viruses, modulate local inflammatory responses, and activate adaptive immune mechanisms (*Banke et al., 2020*; *Ortiz-Casillas et al., 2019*). Most antimicrobial peptides are cationic and target negatively charged bacterial cell membranes, causing lipid bilayer disruption (*Brogden, Ackermann & Huttner, 1997*; *Samael Olascoaga-Del et al., 2018*). Human $\beta$-defensin 1 ($\beta$-defensin) is a 3928.6 Da peptide primarily expressed in epithelia and neutrophils. In addition to its antimicrobial activity, $\beta$-defensin has immunomodulatory effects, being upregulated in various inflammatory conditions (*Schneider et al., 2005*). Patients with biofilm-forming bacteria associated with PJI exhibit considerably higher levels of $\beta$-defensin compared to those without infection (*Fernandez-Torres et al., 2024*). While $\beta$-defensin's role as an antimicrobial agent is well established, its presence in joint fluids from PJI patients has not been thoroughly investigated.

The hypothesis of this study states that certain blood-derived inflammatory markers could serve as an additional diagnostic tool for PJI when used alongside other parameters, such as synovial $\beta$-defensin levels. Therefore, the objectives were: (a) to determine the efficacy of synovial $\beta$-defensin as a diagnostic marker for PJI; (b) to compare the performance of synovial $\beta$-defensin with ESR, serum CRP and WBC in the diagnosis of PJI; and (c) to establish cut-off values for $\beta$-defensin, ESR, CRP, and WBC in the diagnosis of PJI.

## MATERIALS AND METHODS

### Study setting and participants

This single-center retrospective study was derived from a large project (*Fernández-Torres et al., 2024*), which was approved by the Ethics and Research Committee of the Instituto Nacional de Rehabilitación Luis Guillermo Ibarra Ibarra (INRLGII), under protocol number INR-50/22 and conducted in accordance with the Declaration of Helsinki. Samples were collected from October 2022 to June 2023 period. A total of 105 joint fluid samples from INRLGII revision patients were collected intraoperatively, 71 total hip arthroplasty (TKA) and 34 total knee arthroplasty (TKA) and frozen. According to Musculoskeletal Infection Society (MSIS) criteria and internal INRLGII procedures, 64 patients were defined as positive for PJI and the remaining 41 were negative (aseptic failure). Written informed consent was obtained from all participants. The diagnosis of PJI, requires the presence of one of the following criteria must be met: (a) a sinus tract communicating with the prosthesis; (b) isolation of a pathogen from two separate tissue or fluid samples from the affected joint; or (c) meeting four out of six criteria, including elevated ESR and CRP, elevated synovial fluid WBC count, elevated synovial fluid neutrophil percentage, presence of purulence, microorganism isolation in periprosthetic tissue or fluid culture, and >5 neutrophils per high-power field on histologic analysis of periprosthetic tissue at 400× magnification (Table S1) (*Parvizi et al., 2011*). Men and women >18 years of age were included, and all patients received 500 mg of cephalexin prophylactically before surgery. Exclusion criteria included incomplete laboratory data, concomitant inflammatory conditions (*e.g.,* rheumatoid

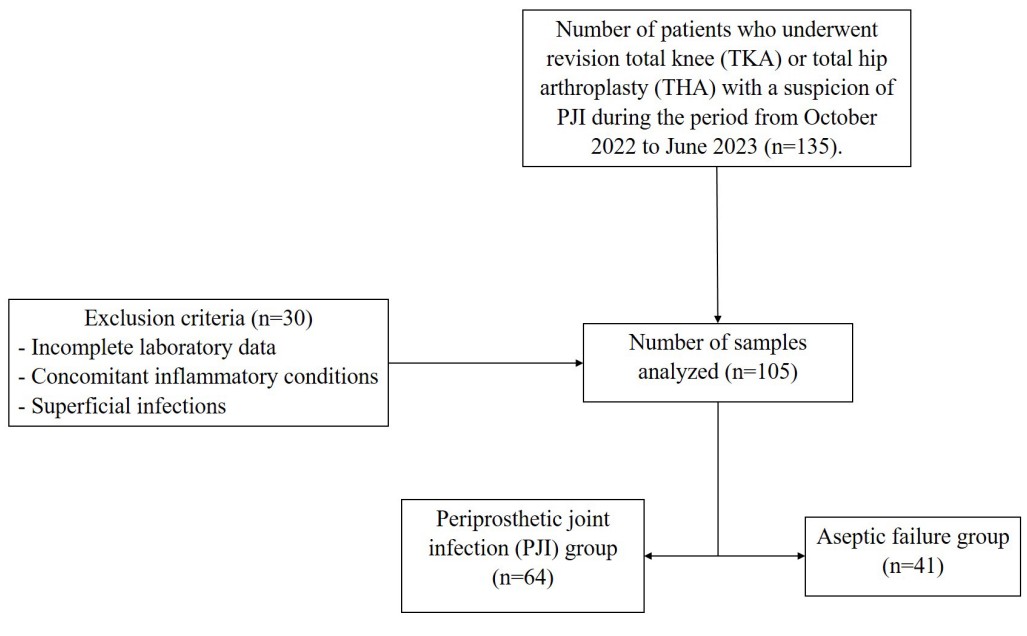

**Figure 1  Flowchart of the study population.**

arthritis, ankylosing spondylitis, microcrystalline arthropathies), superficial infections, a history of malignancy, second-stage two-stage revisions, and periprosthetic fractures, as the inflammation related to these comorbidities could introduce bias into the statistical analysis. Figure 1 shows the flowchart of the study population.

## Sample collection and determination

Samples were collected under aseptic conditions and placed in sterile tubes. Patients' clinical histories were reviewed, and routine diagnostic laboratory tests were performed, including serum CRP, ESR, WBC count, and microbiological culture. Serum CRP levels were quantified by an automated turbidimetric method using Beckman Coulter AU480 equipment (Beckman Coulter, Brea, CA, USA). Serum samples were then frozen at −75 °C until $\beta$-defensin determination.

## Microbiological culture

Samples were processed for aerobes, anaerobes, and fungi and Gram-stained. Biopsies and sonicated samples were inoculated on 5% sheep blood agar and MacConkey agar and incubated under aerobic conditions for 18–24 h. For anaerobes, samples were inoculated on phenylethyl alcohol agar with 5% sheep blood and incubated anaerobically at 37 °C for 48 h. Fungal cultures were grown on dextrose Sabouraud agar supplemented with amikacin and ceftazidime and incubated at room temperature. Biopsies were inoculated into BD BACTEC anaerobic flasks (BD, Franklin Lakes, NJ, USA), sonicated in aerobic and anaerobic flasks, and synovial fluid samples were incubated in pediatric flasks within the BACTEC system for 7 days. Positive samples were further inoculated on 5% blood sheep agar and MacConkey agar and incubated aerobically at 37 °C for 48 h. Anaerobic

recovery was performed on phenylethyl alcohol agar plates with 5% sheep blood, incubated anaerobically at 37 °C for 48 h (excluding synovial fluid). Bacteria and yeasts were identified using the semi-automated Vitek® 2 Compact system (Biomerieux, Marcy-l'Étoile, France).

### Quantification of synovial $\beta$-defensin 1 levels

Synovial $\beta$-defensin levels were quantified using a sandwich-ELISA method with the Human DEFB1 (Defensin Beta 1) ELISA Kit (MBS2500932; MyBioSource) according to the manufacturer's protocol. Optical density was measured at 450 nm using an iMark™ BioRad optical plate reader. Each assay sample was conducted in duplicate, and optical density was converted to concentration (pg/mL) using the standard calibration curve provided by the manufacturer.

### Statistical analysis

Data were analyzed using SPSS 21.0 for Windows® (SPSS Inc., Chicago, IL, USA), with $p$-values <0.05 considered statistically significant. Due to the sample size ($n > 50$), the Kolmogorov–Smirnov test was used to assess the normal distribution of continuous variables. For univariate analysis, the Student's $t$-test was used, or the Wilcoxon signed rank test when appropriate. Demographic data and baseline clinical parameters of patients with PJI were represented as mean ± standard deviation (SD) for quantitative data, while categorical data were described using frequencies and proportions. Variables showing statistical significance in univariate analysis were included in multivariate analysis using a logistic regression model. Sensitivity and specificity values for $\beta$-defensin, ESR, CRP, and WBC were calculated using receiver operating characteristic (ROC) curves, and the area under the curve (AUC) was estimated. Cut-off points were calculated using the Youden index. Positive and negative likelihood ratios (+LR and −LR) were calculated using the Diagnostic Test Calculator tool (freely available at http://araw.mede.uic.edu/cgi-bin/testcalc.pl).

## RESULTS

A total of 105 clinical isolates were collected. The general description of the study population is shown in Table 1. The mean age of participants was 56.6 ± 21.4 years, with 50 women (47.6%) and 55 men (52.4%). Laboratory variables included synovial $\beta$-defensin levels at 1434.1 ± 838.0 pg/mL, ESR at 16.5 ± 13.3 mm/h, serum CRP at 8.6 ± 7.6 mg/L, and WBC at $7.9 ± 2.1 × 10^3/mm^3$. Microbiological culture results were negative in 41 samples (39.1%) and positive in 64 samples (60.9%). The most commonly affected joints with positive cultures were the left hip (34.3%), right hip (33.3%), right knee (21.0%), and left knee (11.4%).

Table 2 summarizes the results by PJI and non-PJI groups. Synovial $\beta$-defensin levels were significantly higher in the PJI group compared to the non-PJI group (1923.0 ± 690.5 *vs.* 671.0 ± 309.2 pg/mL, $p < 0.0001$). Similarly, ESR (22.6 ± 12.5 *vs.* 7.12 ± 8.1 mm/h, $p < 0.0001$), serum CRP (12.6 ± 6.9 *vs.* 2.4 ± 3.2 mg/L, $p < 0.0001$), and WBC ($8.6 ± 2.1$ *vs.* $6.9 ± 1.8 × 10^3/mm^3$, $p < 0.001$) were significantly higher in the septic group. The most commonly isolated pathogens were *Staphylococcus aureus* (*S. aureus*) (28.1%), *Pseudomonas aeruginosa* (*P. aeruginosa*) (28.1%), and *S. epidermidis* (10.9%).

**Table 1  Demographic, laboratory, microbiologic and clinical characteristics of the study population.**

| Parameter | All samples (n = 105) |
|---|---|
| Age (years) ± SD | 56.6 ± 21.4 |
| Sex | |
| Women, n (%) | 50 (47.6) |
| Men, n (%) | 55 (52.4) |
| β-defensin (pg/mL) | 1434.1 ± 838.0 |
| ESR (mm/h) | 16.5 ± 13.3 |
| CRP (mg/L) | 8.6 ± 7.6 |
| WBC ($10^3$/mm$^3$) | 7.9 ± 2.1 |
| Microbiological culture (%) | |
| Positive, n (%) | 64 (60.9) |
| Negative, n (%) | 41 (39.1) |
| Affected joint (%) | |
| Left hip, n (%) | 36 (34.3) |
| Right hip, n (%) | 35 (33.3) |
| Right knee, n (%) | 22 (21.0) |
| Left knee, n (%) | 12 (11.4) |

**Notes.**

Continuous variables are expressed as the mean ± standard deviation (SD).

Abbreviations: ESR, erythrocyte sedimentation rate; CRP, C-reactive protein; WBC, white blood cells.

Variables that were statistically significant in univariate analysis were used as independent variables, with PJI categorized as the dependent variable (No = 0, Yes = 1) for multivariate logistic regression analysis. The results showed that synovial β-defensin (OR = 1.008, 95% CI [1.003–1.012], $p = 0.001$), ESR (OR = 1.163, 95% CI [1.012–1.336], $p = 0.034$), and serum CRP (OR = 1.427, 95% CI [1.071–1.903], $p = 0.015$) were associated with PJI (Table 3).

To evaluate the performance of β-defensin, ESR, CRP, and WBC for PJI, a ROC diagnostic analysis was conducted. The AUCs were 0.948 (0.909–0.987) for synovial β-defensin, 0.884 (0.811–0.956) for ESR, 0.902 (0.842–0.962) for serum CRP, and 0.767 (0.670–0.864) for WBC. The combined AUC for β-defensin, ESR, CRP, and WBC was 0.994 (0.989–0.999). The diagnostic cut-off values were 1105.8 pg/mL for β-defensin, 11.5 mm/h for ESR, 5.5 mg/L for CRP, and $7.3 \times 10^3$/mm$^3$ for WBC (Figs. 2 and 3).

Sensitivity and specificity for the diagnosis of PJI are shown in Table 4. Synovial β-defensin had a sensitivity of 0.966 and specificity of 0.830. For ESR, serum CRP, and WBC, sensitivities and specificities were 0.887 and 0.791, 0.930 and 0.771, and 0.820 and 0.682, respectively. Positive predictive value (PPV) and negative predictive value (NPV) were 0.875 and 0.951 for synovial β-defensin, 0.859 and 0.829 for ESR, 0.828 and 0.902 for serum CRP, and 0.781 and 0.731 for WBC. Positive likelihood ratio (+LR) and negative likelihood ratio (−LR) for β-defensin, ESR, CRP, and WBC were 5.67 and 0.04, 4.24 and 0.14, 4.06 and 0.09, and 2.58 and 0.26, respectively.

**Table 2  Characteristics of the study population by PJI and non-PJI groups.**

|  | PJI (*n* = 64) | Non-PJI (*n* = 41) | *p*-value |
|---|---|---|---|
| Age (years) ± SD | 53.7 ± 21.5 | 61.1 ± 20.6 | 0.083 |
| Sex |  |  |  |
| Women, *n* (%) | 35 (54.7) | 15 (36.6) | 0.076[*] |
| Men, *n* (%) | 29 (45.3) | 26 (63.4) |  |
| *β*-defensin (pg/mL) | 1923.0 ± 690.5 | 671.0 ± 309.2 | **<0.0001** |
| ESR (mm/h) | 22.6 ± 12.5 | 7.12 ± 8.1 | **<0.0001** |
| CRP (mg/L) | 12.6 ± 6.9 | 2.4 ± 3.2 | **<0.0001** |
| WBC ($10^3/mm^3$) | 8.6 ± 2.1 | 6.9 ± 1.8 | **<0.001** |
| Main isolated microorganisms (%) |  |  |  |
| *Staphylococcus aureus*, *n* (%) | 18 (28.1) |  |  |
| *Pseudomonas aeruginosa*, *n* (%) | 18 (28.1) |  |  |
| *Staphylococcus epidermidis*, *n* (%) | 7 (10.9) |  |  |
| *Klebsiella pneumoniae*, *n* (%) | 2 (3.1) |  |  |
| *Staphylococcus haemolyticus*, *n* (%) | 2 (3.1) |  |  |
| *Enterococcus faecalis*, *n* (%) | 1 (1.6) |  |  |
| Others, *n* (%) | 16 (25.0) |  |  |

**Notes.**
The variables are expressed as the mean ± standard deviation (SD).
Abbreviations: PJI, periprosthetic joint infection; ESR, erythrocyte sedimentation rate; CRP, C-reactive protein; WBC, white blood cells.
*p*-values were estimated using *t*-test, $\alpha = 0.05$.
[*]*p*-value was estimated using Fisher's exact test, $\alpha = 0.05$; significant *p*-values are in bold.

**Table 3  Multiple linear regression analysis of factors influencing PJI.**

| Factor | Beta | SE | Wald | OR | 95% CI | *p* |
|---|---|---|---|---|---|---|
| *β*-defensin | 0.008 | 0.002 | 11.517 | 1.008 | 1.003–1.012 | **0.001** |
| ESR | 0.151 | 0.071 | 4.511 | 1.163 | 1.012–1.336 | **0.034** |
| CRP | 0.356 | 0.147 | 5.882 | 1.427 | 1.071–1.903 | **0.015** |
| WBC | 0.030 | 0.302 | 0.010 | 1.031 | 0.571–1.863 | 0.920 |

**Notes.**
Abbreviations: PJI, periprosthetic joint infection; ESR, erythrocyte sedimentation rate; CRP, C-reactive protein; WBC, white blood cells; SE, standard error; OR, odds ratio; CI, confidence interval.
Significant *p*-values are in bold.

## DISCUSSION

*β*-Defensin is produced by various cells, including epithelial, immune, and inflammatory response cells. Factors such as age, sex, and genetics play a crucial role in the host immune response, influencing *β*-defensin levels (*Gameiro & Romao, 2010*; *Berberich, Josse & Ruiz, 2022*; *Alamanda & Springer, 2018*; *Yang et al., 2004*). Surgical site infections, particularly PJI following primary total joint arthroplasty, pose a significant burden by increasing morbidity, mortality, disability, and healthcare costs (*Premkumar et al., 2021*). Gram-positive bacteria are the most commonly isolated pathogens in these cases, presenting a major challenge for healthcare professionals (*Peng et al., 2021*; *Linke et al.,*

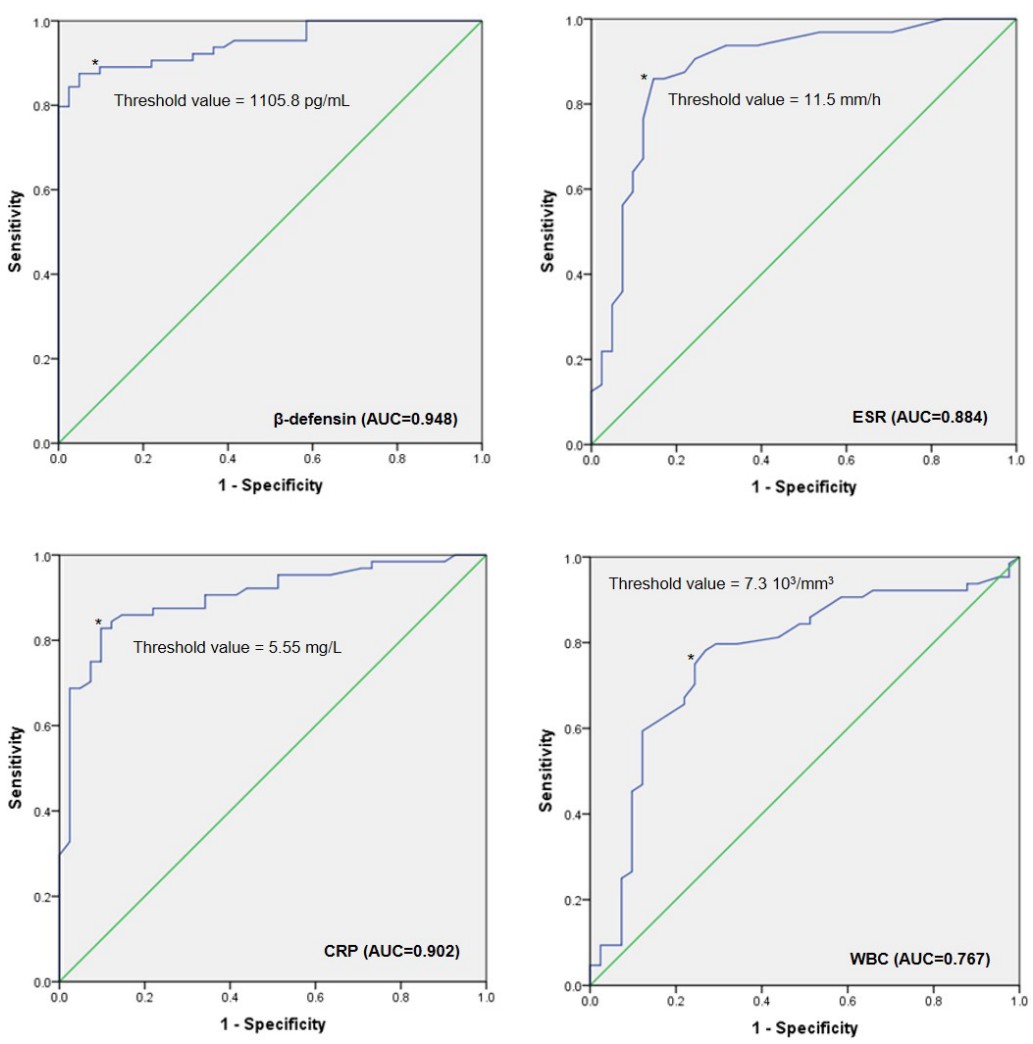

**Figure 2  ROC curves.** The ROC curves illustrate the area under the curve (AUC) values for $\beta$-defensin, ESR, CRP, and WBC, along with their respective cut-off points for diagnosing PJI. Abbreviations: ROC, Receiver operating characteristic; ESR, Erythrocyte sedimentation rate; CRP, C-reactive protein; WBC, White blood cells; PJI, Periprosthetic joint infection. AUC Interpretation: 0.90–1, Excellent; 0.80–0.89, Good; 0.70–0.79, Fair; 0.60–0.69, Poor; 0.50–0.59, Fail.

*2022*). While $\beta$-defensin is constitutively expressed as a natural peptide essential for the innate immune response against various pathogens, its exact mechanism of action remains incompletely understood (*Gollwitzer et al., 2013*; *Ryan & Diamond, 2017*). In our study, the most commonly isolated pathogens were *S. aureus*, *P. aeruginosa*, and *S. epidermidis*. However, we have observed that the levels of $\beta$-defensin or another biomarker may vary depending on the type of microorganism isolated. In a previous work, we analyzed the relationship between synovial $\beta$-defensin, ESR, serum CRP and WBC levels, and we observed that depending on the type of microorganism isolated (*P. aeruginosa* or *S. aureus*), synovial $\beta$-defensin and serum CRP levels were significantly modified compared to the aseptic group ($p < 0.001$ and $p = 0.025$, respectively) (*Fernandez-Torres et al., 2024*). Other

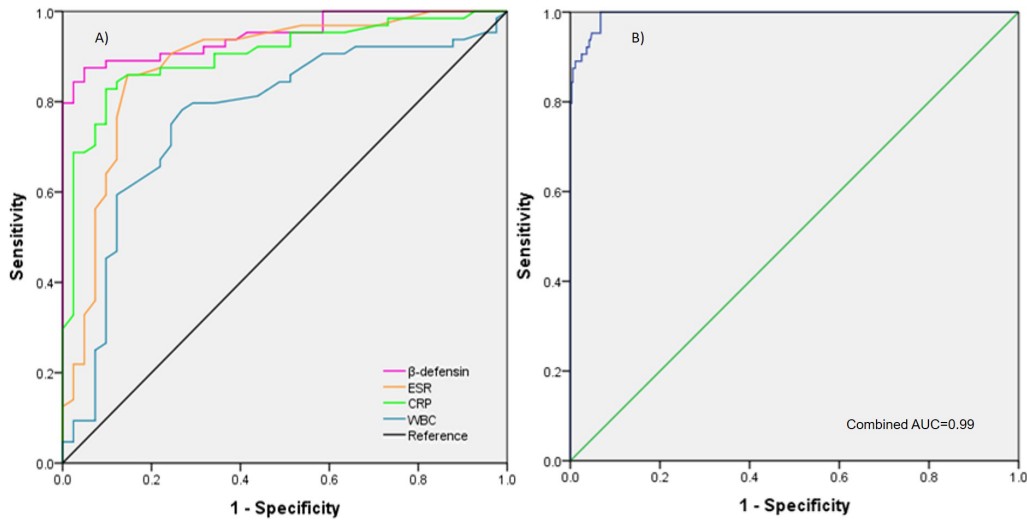

**Figure 3   AUC of each marker and combine.** (A) The AUC values for each of the individual markers ($\beta$-defensin, ESR, CRP, and WBC) shown overlapping on the graph. (B) The AUC when all markers are combined, demonstrating the overall diagnostic performance.

**Table 4   Performance of $\beta$-defensin, ESR, CRP and WBC in the diagnosis of PJI.**

| Biomarker | AUC (95% CI) | p | Cut-off | Sensitivity | Specificity | +LR | −LR | PPV | NPV |
|---|---|---|---|---|---|---|---|---|---|
| $\beta$-defensin | 0.948 (0.909–0.987) | <0.00001 | 1105.8 pg/mL | 0.966 | 0.830 | 5.67 | 0.04 | 0.875 | 0.951 |
| ESR | 0.884 (0.811–0.956) | <0.00001 | 11.5 mm/h | 0.887 | 0.791 | 4.24 | 0.14 | 0.859 | 0.829 |
| CRP | 0.902 (0.842–0.962) | <0.00001 | 5.5 mg/L | 0.930 | 0.771 | 4.06 | 0.09 | 0.828 | 0.902 |
| WBC | 0.767 (0.670–0.864) | 0.00004 | 7.3 $10^3$/mm$^3$ | 0.820 | 0.682 | 2.58 | 0.26 | 0.781 | 0.731 |
| Combined[a] | 0.994 (0.989–0.999) | <0.00001 | − | | | | | | |

**Notes.**
Abbreviations: PJI, periprosthetic joint infection; AUC, area under the curve; ESR, erythrocyte sedimentation rate; CRP, C-reactive protein; WBC, white blood cells; LR, likelihood ratio; PPV, positive predictive value; NPV, negative predictive value.

[a]Combined $\beta$-defensin + ESR + CRP + WBC.

Likelihood Ratio Interpretation:
+LR ≥ 10 and −LR < 0.1, highly relevant; +LR = 5–10 and −LR = 0.1–0.2, Good; +LR = 2–5 and −LR = 0.5–0.2, Regular; +LR < 2 and −LR > 0.5, Poor.

markers may also be modified depending on the type of infection. In *Masters et al. (2022)*, the comparative analysis of infections caused by staphylococci *versus* bacteria other than staphylococci and *S. aureus versus S. epidermidis* showed increased expression of IL-13, IL-17D, and MMP3 in staphylococcal infections; and IL-1 $\beta$, IL8, and platelet factor PF4V1 in *S. aureus* infections compared to *S. epidermidis* infections.

Joint biopsy is a useful tool for diagnosing PJI but has the disadvantages of being invasive and risking contamination of a previously aseptic joint. Inflammatory markers such as IL-6, ESR, CRP, and WBC count are commonly evaluated in suspected PJI cases; however, these markers can be elevated due to other causes like concomitant infections, systemic inflammatory diseases, and local conditions such as rheumatoid arthritis and gout, thus lacking specificity (*Di Cesare et al., 2005*; *Archibeck et al., 2001*). *Tetreault et al.*

*(2014)* compared serum and synovial fluid CRP measurements and found that synovial fluid CRP did not provide a diagnostic advantage over serum CRP in detecting PJI.

In this study, we evaluated the performance of $\beta$-defensin levels in joint fluids as a potential diagnostic marker. Our results demonstrated that synovial $\beta$-defensin had higher sensitivity and specificity than ESR, serum CRP, and WBC. Synovial $\beta$-defensin, ESR, and serum CRP were significantly associated with PJI. To our knowledge, there are no previous reports on using synovial $\beta$-defensin as a diagnostic biomarker for PJI. But other molecules or methods with high sensitivity and specificity have been tested. For instance, $\alpha$-defensin has shown high sensitivity and specificity (~100% and 95%, respectively) (*Frangiamore et al., 2016*; *Bingham et al., 2014*). It is worth mentioning that when $\alpha$-defensin is used in combination with leukocyte esterase, the sensitivity and specificity for the diagnosis of PJI are further increased (*Li et al., 2020*). However, it should be noted that the sensitivity and specificity are usually not as effective in shoulder PJI as in TKA and THA (*Unter Ecker et al., 2019*). It is worth noting that defensins are classified into the $\alpha$, $\beta$, and $\theta$ subfamilies according to the position of their disulfide bonds; however, only the $\alpha$ and $\beta$ defensins are found in humans (*Zhai et al., 2023*) These subtypes differ in the length of their amino acid residues, the location of the disulfide bonds, and the cell type in which they are expressed. While leukocyte esterase is an enzyme secreted by activated neutrophils and macrophages that helps degrade molecules on bacteria and other pathogens during an immune response, it has been widely used as an indicator to assess urinary tract infections in the clinic (*Li et al., 2020*). Therefore, due to the paucity of assays that simultaneously compare the performance of $\alpha$-defensin, $\beta$-defensin, and leukocyte esterase in the diagnosis of PJI, it remains open for future studies of this type.

Nonetheless, some studies suggest biopsy as the first line of diagnosis for PJI. *Fink et al. (2013)* found the biopsy technique superior to aspiration and CRP in diagnosing hip PJI, especially in patients with negative aspirates but elevated CRP or clinical signs of infection, where biopsy was preferable to repeat aspiration. In contrast, *Williams, Norman & Stockley (2004)* argued that tissue punch biopsy is more invasive and offers no advantage over aspiration in terms of bacterial accuracy, often resulting in more false-positive results.

*Froschen et al. (2020)* evaluated a panel of individual cytokines such as IL-6, IL-1 $\beta$, IL-10, and IL-17 in synovial fluid and found high sensitivity and specificity values for diagnosing PJI when comparing septic and non-septic groups. This panel could be a useful predictive tool for determining the likelihood ratio of PJI in patients with a painful prosthesis. The use of $\beta$-defensin as a marker in PJI is based on its ability to indicate the presence of infection. Specifically, measuring $\beta$-defensin levels in joint fluids or biopsies can help differentiate between infection and a non-infectious inflammatory response associated with a prosthetic joint.

This study demonstrated that synovial $\beta$-defensin may have the potential to be used as a reliable biomarker for the diagnosis of PJI. However, several limitations were identified. First, the single-center retrospective design may have introduced a selection bias affecting the results, so the results should be interpreted with caution. This situation can be corrected by developing prospective multicenter studies. Second, our design was based on the MSIS criteria, and currently these definitions have been modified as those of the European Bone

and Joint Infection Society (EBJIS) of 2021. Finally, there is a risk of sample contamination after aspiration, which could lead to false positives, and cultures for anaerobes had to be maintained for up to 14 days before being declared negative, as *Cutibacterium acnes* is a slow-growing biofilm-forming bacterium and is also one of the most common etiologic agents in PJI.

## CONCLUSION

In summary, our results demonstrated that the synovial $\beta$-defensin assay has higher sensitivity and specificity compared to ESR, serum CRP, and WBC, suggesting its significant potential as a diagnostic marker for PJI. Although not a perfect test, $\beta$-defensin could be considered a valuable tool within the existing diagnostic criteria for PJI. Moreover, the simultaneous determination of multiple markers may enhance diagnostic confidence.

### Funding
The authors received no funding for this work.

### Competing Interests
The authors declare there are no competing interests.

### Author Contributions
- Javier Fernández-Torres conceived and designed the experiments, performed the experiments, analyzed the data, prepared figures and/or tables, authored or reviewed drafts of the article, and approved the final draft.
- Yessica Zamudio Cuevas conceived and designed the experiments, performed the experiments, analyzed the data, prepared figures and/or tables, authored or reviewed drafts of the article, and approved the final draft.
- Karina Martínez Flores performed the experiments, analyzed the data, prepared figures and/or tables, authored or reviewed drafts of the article, and approved the final draft.
- Ambar López Macay performed the experiments, authored or reviewed drafts of the article, and approved the final draft.
- Graciela Rosas Alquicira performed the experiments, authored or reviewed drafts of the article, and approved the final draft.
- María Guadalupe Martínez-Zavaleta performed the experiments, authored or reviewed drafts of the article, and approved the final draft.
- Luis Esaú López Jácome performed the experiments, authored or reviewed drafts of the article, and approved the final draft.
- Rafael Franco-Cendejas conceived and designed the experiments, authored or reviewed drafts of the article, and approved the final draft.
- Ernesto Roldan-Valadez analyzed the data, authored or reviewed drafts of the article, and approved the final draft.

## Human Ethics

The following information was supplied relating to ethical approvals (*i.e.,* approving body and any reference numbers):

The Instituto Nacional de Rehabilitación Luis Guillermo Ibarra Ibarra granted Ethical approval to carry out the study within its facilities (Ref: 50/22).

## Data Availability

The raw measurements are available in the Data S1. Raw data includes clinical, anthropometric and biochemical variables.

## Supplemental Information

Supplemental information for this article can be found online at http://dx.doi.org/10.7717/peerj.18560#supplemental-information.

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
