# Peer review of "β-Defensin *versus* conventional markers of inflammation in periprosthetic joint infection: a retrospective study"

_PeerJ, doi:10.7717/peerj.18560_

## Round 0.1 · original submission · Major Revisions

Please address issues pointed by both reviewers and amend manuscript accordingly.

·

Basic reporting

This study evaluates the usefulness of β-defensin in diagnosing PJI from 105 joint fluid samples. The authors suggest that β-defensin is more accurate in diagnosing infections compared to CRP, ESR, and WBC. The β-defensin used in this study is a peptide mainly released by immune cells when bacteria are identified, and various studies have demonstrated that identifying these antimicrobial peptides is valuable for diagnosing PJI.

Experimental design

The most important aspect of this study is whether the diagnostic accuracy of β-defensin is superior in terms of precision, cost-effectiveness, simplicity, and versatility compared to joint fluid tests included in PJI diagnostic criteria, such as neutrophil esterase or α-defensin. However, such comparisons were not achieved in this study. Since the roles of blood tests and joint fluid tests differ significantly, I strongly recommend to author to conduct the comparison with synovial fluid test to fully evaluate the usefulness of synovial β-defensin.

Validity of the findings

no comment

Additional comments

no comment

Reviewer 2 ·

Basic reporting

no comment

Experimental design

no comment

Validity of the findings

no comment

Additional comments

The research aim is to determine the value of beta-defensin as a useful marker for PJI diagnosis. The abstract is structured; the keywords should be checked in accordance to MeSH.
The introduction transposes the research into the topic and formulates the objective of the study at the end. However, this should be addressed in a from of hypothesis. Also, more examples of inflammatory markers used for PJI diagnosis should be given in relation tot the scientific literature (for e.g. doi: 10.3390/jcm13195716).
In the methodology section, the stages of the research are presented in an organized manner. Why were the MISIS criteria used when the current practice used the EBJIS criteria? (this should be explained by the authors). The exclusion criteria is poorly defined, there are a lot of conditions that can influence the results such as ALTR, metastases etc… The administration of antibiotic before surgery and during sampling is mandatory (were patients on septicemia or were they under antibiotic treatment?). In the “Statistical analysis” subsection please provide the normality check test used and also the statistical tests perfumed in the univariate analysis.
In the results section more in depth statistical analysis could be performed sub as the relation of beta-defensin and the type of microorganisms. A flowchart of the studied population could be added.
The discussions interpret the research results and relate them to other results from scientific literature. The authors should make a comparing with alpha-defensin as it is more widely used. Limitations of the study should be provided more clear at the end of the section in a well defined paragraph
The conclusions are concise.
The references are adequate but need proper editing and should be extended as suggested above.

---

## Round 0.2 · Minor Revisions

Both reviewers were mostly satisfied by the revision. However, reviewer #1 indicated that there are two additional minor clarifications that need to be included to the manuscript.

·

Basic reporting

Thank you for providing appropriate responses to the reviewer’s questions and for revising the manuscript. I have only two additional requests, which I would appreciate if you could incorporate into the manuscript:
1. Currently, β-defensin is listed alongside CRP, ESR, and D-dimer. To help clarify the distinction between synovial fluid tests and blood tests, I suggest specifying terms like "synovial β-defensin" and "serum CRP."
2. It would be helpful to briefly explain the differences between β-defensin and α-defensin, as well as leukocyte esterase.

Experimental design

No comment

Validity of the findings

No comment

Additional comments

No comment

Reviewer 2 ·

Basic reporting

-

Experimental design

-

Validity of the findings

-

Additional comments

The authors have improved their paper accordingly and it is now suitable for publication.

---

## Round 0.3 · accepted · Accept

All the remaining issues were adequately addressed and revised manuscript is acceptable now.